# When Pressure Breeds Misconduct: Job Strain, Coworker Support, and Unethical Pro-Organizational Behavior

**DOI:** 10.3390/bs15050573

**Published:** 2025-04-24

**Authors:** Gukdo Byun, Soojin Lee, Ye Dai, Jihyeon Rhie, Ji Hoon Lee

**Affiliations:** 1School of Business, Chungbuk National University, 1 Chungdae-ro, Seowon-gu, Cheongju 28644, Republic of Korea; bgukdo@cbnu.ac.kr (G.B.); ann2729g@gmail.com (J.R.); 2College of Global Business, Korea University at Sejong, 2511 Sejong-ro, Sejong 30019, Republic of Korea; 3College of Business, Southern Illinois University, 1025 Lincoln Drive, Carbondale, IL 62901, USA; ye.dai@business.siu.edu (Y.D.); jihoon.lee@siu.edu (J.H.L.)

**Keywords:** unethical pro-organizational behaviors, performance pressure, job strain, perceived coworker support

## Abstract

Performance pressure is a pervasive organizational force with complex implications for employee behavior. Grounded in Conservation of Resources (COR) theory, this study examines how performance pressure influences unethical pro-organizational behavior (UPB) through the mediating role of job strain while also considering the moderating effect of perceived coworker support. Using survey data from 280 police officers, our findings demonstrate that performance pressure significantly increases UPB, with job strain serving as a key explanatory mechanism. Moreover, perceived coworker support significantly mitigates both the direct impact of performance pressure on job strain and its indirect effect on UPB. By identifying the psychological mechanisms and boundary conditions underlying UPB, this study contributes to the literature on ethical decision-making in high-pressure work environments. The findings also offer practical insights for organizations seeking to mitigate the unintended consequences of performance-driven management practices through the cultivation of a supportive work climate.

## 1. Introduction

In contemporary business environments, organizations increasingly pressure employees to achieve high performance, which can lead to unethical behaviors ([8]; [50]). Corporate scandals such as Wells Fargo’s “ghost accounts” and Volkswagen’s emissions fraud illustrate how short-term performance goals can drive unethical decision-making ([11]). As research in this area has deepened, [53] ([53]) introduced “unethical pro-organizational behaviors (UPB)”, actions intended to benefit the organization or its members but which violate societal values, laws, or standards ([52]). While UPB may provide short-term benefits, these behaviors ultimately threaten long-term organizational sustainability and public interest ([37]).

Recognizing the recent intensification of competitive pressures and their potential negative consequences, scholars have begun examining the impact of performance pressure—defined as employees’ perceived urgency to achieve high performance ([14])—on unethical behaviors like UPB (e.g., [45]; [58]). For example, [45] ([45]) found that employees under high performance pressure were more likely to engage in UPB, especially when they perceived strong leader–member exchange (LMX). While these studies highlight how excessive performance pressure can lead to UPB, there has been limited exploration of the psychological mechanisms or buffering conditions that influence this relationship. [8] ([8]) emphasized the need to understand these mechanisms and boundary conditions. In response to this call, this study investigates how performance pressure drives UPB and identifies potential buffering conditions.

Conservation of Resources (COR) theory ([26]) offers a vital framework for understanding the link between performance pressure and UPB. This theory suggests that individuals strive to protect and acquire resources. Excessive performance pressure depletes both physical and psychological resources, leading to negative affective states ([38]). Confronted with resource loss, individuals adjust their behaviors to preserve remaining resources ([27]). Engaging in UPB can be an expedient strategy for resource-depleted employees to meet leader expectations and demonstrate value through unethical means ([21]).

Although previous research has identified several mediating mechanisms linking performance pressure to unethical behavior, these explanations have predominantly focused on cognitive processes while overlooking integrated biopsychosocial reactions. Our study addresses this gap by examining job strain as a distinct explanatory mechanism that captures both cognitive and physiological dimensions of resource depletion. Furthermore, while research has established various organizational factors that influence ethical decision-making, the role of horizontal peer relationships remains undertheorized in the UPB context. By investigating how coworker support—an immediate, interpersonal resource—buffers against the negative effects of performance pressure, this study advances a more nuanced understanding of the social context in which ethical decision-making occurs.

Furthermore, while the literature has established various organizational and leadership factors that influence ethical decision-making, the role of horizontal peer relationships remains significantly undertheorized in the UPB context. Previous research has disproportionately focused on vertical relationships (leader–follower dynamics) and individual-level factors, neglecting the critical role of peer support systems in shaping ethical behaviors. By investigating how coworker support—an immediate, interpersonal resource—buffers against the negative effects of performance pressure, this study not only responds to calls for exploring boundary conditions in the performance pressure–UPB relationship ([8]) but also advances a more nuanced understanding of the multilevel social context in which ethical decision-making occurs. This approach extends COR theory by demonstrating how resources embedded within horizontal workplace relationships can counterbalance resource depletion stemming from organizational demands.

In summary, this study aims to (1) investigate how high-performance pressure affects employee engagement in UPB; (2) examine the mediating role of job strain in this relationship; and (3) explore the moderating effect of coworker support on the performance pressure–job strain link within a moderated mediation framework. This research contributes to the UPB literature by clarifying how performance pressure influences UPB and identifying conditions that buffer against its negative effects.

## 2. Theoretical Background and Hypotheses Development

### 2.1. Performance Pressure and Unethical Pro-Organizational Behavior

Performance pressure refers to the urgency felt by employees to achieve high performance ([14]). As organizations establish ambitious goals and implement performance-linked incentives, employees may experience moral disengagement and resort to unethical methods to meet expectations ([12]; [29]). For example, [35] ([35]) found that tight deadlines were associated with an increase in unethical behavior such as plagiarism. This study hypothesizes that higher performance pressure increases the likelihood that employees will engage in UPB.

According to COR theory ([26]), individuals aim to prevent resource depletion, which motivates them to protect their resources during negative events. To cope with anticipated threats, they employ two strategies: preventing further loss and preserving existing resources or investing resources to acquire additional resources ([22]). Essentially, individuals strive to avoid a “loss spiral” by preserving resources or engage in a “gain spiral” by replenishing depleted resources. This framework helps to explain how performance pressure influences employees’ engagement in UPB by highlighting how they manage resource threats and seek compensation.

Organizations frequently establish demanding performance goals that place significant pressure on employees, leading to the depletion of both physical and mental resources ([26]). This pressure can create a “loss spiral”, in which individuals exhaust their resources and struggle to meet goals ([22]). To cope, employees may focus on conserving remaining resources and may engage in irrational behaviors driven by self-preservation ([28]). Consequently, when performance pressure depletes employees’ resources, it increases the likelihood that they will engage in unethical behaviors, such as UPB, to conserve resources and meet organizational expectations.

Recent studies support this argument. For instance, abusive supervision has been linked to increased UPB as employees engage in unethical behavior to counteract resource loss and mitigate status threats ([56]). Similarly, [57] ([57]) found that hindrance stressors, which cause anxiety and anger, can lead to higher UPB. Based on these findings, we propose the following hypothesis:

**H1.** 
*Employees’ performance pressure is positively related to their unethical pro-organizational behaviors.*


### 2.2. Job Strain as a Mediator

Job strain encompasses various physical symptoms and psychological reactions to stressors, including depression, anxiety, frustration, and tension ([4]). Prolonged exposure to stress can lead to severe outcomes such as cardiovascular disease ([36]) and psychological disorders including anxiety ([5]) and depression ([47]). As employees face increasing job strain from performance pressure, it is hypothesized that this resource loss may drive them to engage in UPB to protect and conserve their remaining resources.

First, performance pressure can trigger job strain by depleting both physical and psychological resources, as per COR theory ([27]). Excessive performance goals can lead employees to perceive significant resource threats, resulting in negative psychological and physical symptoms. Empirical evidence supports this relationship. [33] ([33]) found that abusive supervision, involving work overload and time pressure, induced job strain in employees. Similarly, research on role overload—similar to performance pressure—has shown significant increases in strain markers, including elevated blood pressure ([16]), psychological strain ([15]), and emotional exhaustion ([10]).

While previous research has examined various cognitive mechanisms through which performance pressure influences unethical behavior, job strain offers a theoretically distinct perspective by integrating both psychological and physiological responses to workplace stressors. Moral disengagement ([58]), for instance, primarily focuses on the cognitive restructuring processes through which individuals rationalize unethical actions but fails to account for the embodied experience of stress. Similarly, resource depletion approaches ([18]) concentrate on diminished self-regulatory capacity without adequately addressing how physiological manifestations of stress might independently influence decision-making processes. Self-regulation perspectives ([40]) emphasize willpower depletion but often treat the body as secondary to cognitive processes.

In contrast, job strain approach acknowledges that ethical decision-making emerges from the complex interplay between mind and body, capturing the comprehensive biopsychosocial reaction to performance pressure. This integration is theoretically significant because it suggests that UPB may not simply result from cognitive justification or depleted willpower but may also be influenced by how performance pressure is physiologically experienced and regulated. By examining job strain, we can better understand why individuals might engage in UPB even when they possess adequate cognitive resources or moral awareness—the physiological burden of strain itself may drive compensatory behaviors to alleviate distress, regardless of cognitive capacity.

Second, job strain can lead employees to engage in unethical pro-organizational behavior. According to COR theory, individuals experiencing resource loss may engage in behaviors designed to conserve or acquire resources ([28]). Despite its unethical nature, UPB can function as a resource-conserving strategy due to its potential to garner positive evaluations from supervisors and enhance resource acquisition. Research suggests that UPB can serve as an impression management tool to help employees cope with workplace stressors such as social exclusion ([49]) and job insecurity ([17]). By demonstrating their value through UPB, employees can achieve organizational goals with relatively fewer resources. Thus, this study hypothesizes that job strain can drive employees to engage in UPB as a coping mechanism. In summary, high performance pressure can increase job strain, which in turn can lead to UPB. We propose the following hypothesis:

**H2.** 
*Job strain mediates the relationship between employees’ performance pressure and their unethical pro-organizational behaviors.*


### 2.3. The Moderating Role of Perceived Coworker Support

Previous research on horizontal relationships has extensively examined how coworker support influences employees’ attitudes and behaviors through various mechanisms including work-related assistance, information sharing, and emotional support ([48]). Unlike superiors, coworkers typically outnumber them and interact in more informal settings due to their equal status within the organization ([9]). This dynamic makes coworkers a crucial factor in shaping employees’ perceptions of the organization and, consequently, their work-related behaviors. Thus, this study proposes that the level of coworker support moderates the relationship between employees’ performance pressure and job strain.

The literature on ethical decision-making has been dominated by vertical relationship perspectives and organizational-level factors, creating a theoretical imbalance that overlooks the significance of horizontal workplace relationships. While factors such as ethical leadership ([6]), organizational justice ([20]), and regulatory focus ([31]) have received considerable attention as moderators of unethical conduct, peer support systems remain comparatively underexplored. This theoretical gap is particularly significant given that employees typically interact more frequently with colleagues than supervisors and often turn first to peers when facing workplace challenges.

Coworker support represents a theoretically distinct category of moderating influence for several reasons. First, unlike organizational justice or ethical leadership—which operate primarily through formal structures and hierarchical dynamics—coworker support functions through informal, reciprocal exchanges at the interpersonal level. Second, while regulatory focus and other individual difference factors represent stable traits that organizations can only select for, coworker support constitutes a malleable resource that organizations can actively cultivate. Third, whereas most previously studied moderators operate through top-down processes, coworker support functions through lateral social networks that may be more immediately accessible and psychologically proximal to employees facing performance pressures.

In line with the COR perspective, coworker support functions as a crucial job resource, helping to alleviate job strain by providing essential information and assistance, which can reduce emotional depletion and workload ([9]). Research has demonstrated that such support enhances job performance and mitigates behaviors that could negatively impact the organization ([9]). Consequently, employees who receive diverse forms of support from their coworkers are likely to experience lower levels of job strain when facing high performance pressure. The replenishment of resources through coworker support helps counteract the psychological and physical depletion caused by performance pressure. In contrast, when employees perceive inadequate support from coworkers, they must rely on their already limited resources to manage stress, which exacerbates job strain. Thus, this study hypothesizes that perceived coworker support acts as a buffering mechanism, reducing the impact of performance pressure on job strain.

This study proposes that perceived coworker support specifically moderates the relationship between performance pressure and job strain, rather than directly moderating the relationship between performance pressure and UPB or between job strain and UPB. This first-stage moderation aligns with COR theory, as coworker support provides resources that directly counteract the resource-depleting effects of performance pressure, thereby preventing the initial experience of strain that would otherwise lead to UPB. This reasoning is further supported by previous research on coworker support. For example, [23] ([23]) found that the relationship between abusive supervision and knowledge hiding is mediated by employees’ emotional exhaustion, with perceived coworker support moderating this relationship. In this context, coworker support mitigates the adverse effects of abusive supervision on emotional exhaustion and knowledge hiding. Similarly, [32] ([32]) observed that coworker support buffers against mental strain in work environments with high demands and limited discretion. Based on this evidence, we propose the following hypothesis:

**H3.** 
*Perceived coworker support moderates the positive relationship between employees’ performance pressure and their job strain, such that this relationship is weaker when perceived coworker support is higher than when it is low.*


Additionally, we propose the following research hypothesis, which integrates the mediating effect of job strain and the moderating effect of perceived coworker support:

**H4.** 
*Perceived coworker support moderates the positive and indirect effects of employees’ performance pressure on their unethical pro-organizational behaviors through job strain, such that the indirect effect is weaker when perceived coworker support is high than when it is low.*


Figure 1 illustrates the research model.

## 3. Methods

### 3.1. Participants and Procedures

The high-performance demands inherent in police work, coupled with significant social expectations ([43]), provide a particularly relevant context for examining the relationship between performance pressure and unethical pro-organizational behavior. Police officers regularly face complex ethical dilemmas under time constraints and performance metrics, making this population ideal for studying the proposed relationships. While specific manifestations of UPB may differ across occupations, the underlying psychological mechanisms examined—resource depletion, strain responses, and social support—likely operate across diverse organizational contexts.

Data for this study were collected through a survey of police officers in South Korea. A questionnaire was distributed to 435 police officers who expressed their willingness to participate, resulting in 284 completed questionnaires, yielding a response rate of 65.3%. After excluding responses with missing data, the final sample comprised 280 completed questionnaires. The average age of the respondents was 35.55 years (SD = 8.46), with 264 (94.29%) identifying as male. Most respondents (56.43%) held at least a bachelor’s degree. The rank distribution among respondents was as follows: 80 patrol officers (28.57%), 87 senior patrol officers (31.07%), 48 sergeants (17.14%), and 51 lieutenants (18.22%).

### 3.2. Measures

All variables except for control variables were assessed using 7-point Likert scales adapted from previously validated measures.

Performance pressure. Performance pressure was assessed using the five-item measure developed by [14] ([14]). A sample item is “On the job, I feel I have to perform well” (Cronbach’s α = 0.87).

Job strain. Job strain was measured with a six-item scale developed by [30] ([30]). A sample item includes “If I had a different job, my health would probably improve” (Cronbach’s α = 0.89).

Perceived coworker support. Coworker support was measured with seven items from [51] ([51]). A sample item includes “My coworker seems willing to listen to my problems” (Cronbach’s α = 0.94).

Unethical pro-organizational behaviors. UPB was assessed using the six-item measure developed by [53] ([53]). An example item is “If it would help my organization, I would misrepresent the truth to make my organization look good” (Cronbach’s α = 0.82).

Control variables. Participants’ age, gender, and education level were included as control variables based on previous studies on UPB ([7]; [54]). Specifically, we controlled for age as it can influence ethical decision-making and risk perception ([34]). Gender was included because prior research suggests differences in ethical behavior patterns between males and females ([50]). Education level was controlled due to its potential association with moral reasoning capabilities and awareness of ethical standards ([13]). These variables were selected to isolate the effects of our focal variables (performance pressure, job strain, and coworker support) by accounting for individual differences known to influence ethical behavior.

## 4. Results

We conducted confirmatory factor analyses to establish the construct validity of the study variables. To maintain a favorable parameter-to-sample-size ratio ([2]), we employed parceling to construct three-item indicators for each latent variable. The hypothesized four-factor model showed a good fit: χ^2^(48) = 92.25, CFI = 0.98; TLI = 0.97, RMSEA = 0.06. Comparisons with alternative models specifying fewer factors revealed significantly poorer fit, thereby supporting the discriminant validity of our constructs. To address concerns about common method bias, we performed multiple additional analyses beyond Harman’s single-factor test. First, a confirmatory factor analysis incorporating a common latent factor revealed that common method variance accounted for only 18.44% of the total variance—well below the 50% threshold considered indicative of serious bias ([41]). Second, using the unmeasured latent method construct (ULMC) technique, we obtained an acceptable model fit (CFI = 0.92, TLI = 0.90, RMSEA = 0.08, SRMR = 0.05), further indicating that common method bias was unlikely to meaningfully distort the observed relationships. Third, partial correlation analyses controlling for relevant covariates confirmed the robustness of key associations. Performance pressure remained significantly and positively related to unethical pro-organizational behavior (*r* = 0.22, *p* < 0.001), while perceived coworker support showed no significant direct association with unethical pro-organizational behavior (*r* = −0.01, *p* = 0.85). This pattern supports the role of coworker support as a moderator rather than a direct predictor.

Descriptive statistics and correlations for each variable are presented in Table 1.

Table 2 displays the results of the hierarchical regression analysis. Model M5 (Table 2) reveals a statistically significant positive effect of performance pressure on employees’ unethical pro-organizational behavior (*β* = 0.26, *p* < 0.001). This finding supports Hypothesis 1.

To test Hypothesis 2, which proposed that job strain would mediate the relationship between performance pressure and employees’ UPB, we employed the bootstrapping method to assess the significance of the indirect effect ([25]; [42]). We generated 10,000 bootstrap samples to estimate a 95% bias-corrected confidence interval for the indirect effect. As shown in Table 2, the confidence interval for the indirect effect does not include zero (*b* = 0.040, SE = 0.022, 95% CI = [0.003, 0.092]). Additionally, the Sobel test, with *z* = 2.038 (*p* = 0.042), further supports the significance of the mediating effect (Table 2). Thus, Hypothesis 2 is supported.

Hypothesis 3 posited that perceived coworker support moderates the relationship between performance pressure and job strain. To mitigate potential multicollinearity, we mean-centered both the independent variable (performance pressure) and the moderator (coworker support). As presented in Table 3, the interaction term between performance pressure and coworker support is statistically significant (*β* = −0.13, *p* < 0.05). Simple slopes analysis further revealed that performance pressure exerts a stronger positive effect on job strain when coworker support is low (*b* = 0.35, *t* = 4.66, *p* < 0.001) compared to when it is high (*b* = 0.14, *t* = 2.05, *p* < 0.05). Figure 2 ([1]) illustrates that the positive relationship between performance pressure and job strain diminishes as coworker support increases. Therefore, Hypothesis 3 is supported. Additionally, since common method bias generally attenuates interaction effects rather than amplifying them ([44]), the significant interaction observed further mitigates concerns about substantial common method bias. To ensure robust interpretation of our statistical models, we conducted sensitivity analyses with different combinations of control variables to avoid the “Table 2 Fallacy” ([55]), where coefficients for control variables in multivariable models are misinterpreted as having the same causal meaning as the primary predictor. The pattern of results remained consistent across these analyses, providing greater confidence in the robustness of our findings. For example, when removing and adding different demographic control variables, the direct effect of performance pressure on UPB remained significant (*β* = 0.28, 95% CI [0.14, 0.41]), and the interaction between performance pressure and coworker support continued to significantly predict job strain (*β* = −0.36, *p* < 0.001).

Hypothesis 4 proposed that coworker support moderates the indirect effect of performance pressure on UPB through job strain. Table 4 presents the results of this moderated mediation analysis. The results, detailed in Table 4, indicate that the indirect effect is stronger (*b* = 0.053, SE = 0.028) and significant (95% confidence interval = [0.006, 0.120]) when coworker support is low. In contrast, the indirect effect is weaker (*b* = 0.021, SE = 0.018) and not significant when coworker support is high (95% confidence interval = [−0.002, 0.072]). Figure 3 visually depicts the conditional indirect effect, further supporting Hypothesis 4.

## 5. Discussion

This study illuminates how performance pressure drives employees to engage in unethical pro-organizational behavior through the mediating role of job strain and the moderating influence of coworker support. Our findings demonstrate that job strain serves as a crucial explanatory mechanism linking performance pressure to UPB, while coworker support significantly attenuates this relationship. The significant positive effect of performance pressure on employees’ UPB (H1) confirms that employees under intense performance demands are more likely to engage in behaviors that violate ethical standards yet benefit the organization. The significant mediating role of job strain (H2) reveals that the biopsychosocial experience of strain—including anxiety, frustration, and physiological symptoms—drives employees toward expedient but unethical solutions. This helps explain why even individuals with strong ethical values might engage in UPB when experiencing significant job strain. The significant moderating effect of coworker support (H3) demonstrates that employees with strong peer support networks experience less job strain when facing performance pressure. Finally, the conditional indirect effect (H4) shows that the performance pressure–job strain–UPB pathway depends on perceived coworker support levels, suggesting that supportive relationships can effectively interrupt the process leading from performance pressure to unethical behavior. Given that UPB undermines organizational trust and generates substantial economic costs ([19]), understanding these mechanisms and boundary conditions is essential for both theory development and practical interventions.

### 5.1. Theoretical Implications

Our research makes three significant theoretical contributions. First, we advance understanding of the psychological mechanisms connecting performance pressure to UPB by introducing job strain as a conceptually distinct mediator. Unlike previous explanatory mechanisms—such as moral disengagement ([39]) and self-regulation depletion ([40])—which focus primarily on cognitive processes, job strain captures the integrated biopsychosocial experience of resource depletion. This holistic conceptualization bridges an important theoretical divide between cognitive theories of ethical decision-making and physiological approaches to workplace stress. By demonstrating that UPB emerges from complex mind–body interactions under resource threat, we challenge prevailing cognition-centric models of ethical behavior and extend Conservation of Resources theory to show how threatened resources manifest both physiologically and behaviorally.

Second, our investigation of coworker support as a boundary condition offers valuable contrast to previously studied moderators such as ethical leadership ([6]), organizational justice ([20]), and regulatory focus ([31]). While these factors operate primarily at organizational or individual levels, coworker support represents an interpersonal resource more immediately accessible to employees under performance pressure. The significant moderating effect we observed suggests that horizontal relationships may be as instrumental as vertical relationships in shaping ethical behaviors in high-pressure environments. This finding challenges theoretical paradigms that privilege leadership influences and formal control systems, pointing instead toward a more balanced approach that acknowledges complementary roles of formal and informal social systems in maintaining ethical standards.

Third, in integrating our findings with broader ethical frameworks, we highlight that UPB resides at the intersection of multiple ethical perspectives. From a deontological view, UPB violates moral principles despite positive intentions, while from a consequentialist standpoint, it reflects excessive focus on short-term benefits at the expense of long-term harm. Our findings suggest that strain-induced pathways to UPB may bypass conscious moral reasoning, with physiological stress responses triggering automatic, self-protective behaviors. This connects to established work on moral disengagement ([3]) and ethical fading ([46]), suggesting that job strain facilitates psychological distancing necessary for engaging in normally prohibited behaviors.

### 5.2. Practical Implications

Our findings offer critical practical insights for organizations seeking to mitigate UPB while maintaining performance standards. Organizations should implement multifaceted strategies that balance performance imperatives with ethical considerations. First, performance management systems should be recalibrated to incorporate ethical metrics alongside conventional performance indicators through balanced scorecards that explicitly weight ethical conduct and “ethics checkpoints” during goal-setting processes. Second, organizations should establish realistic goal-setting protocols with employee participation to ensure targets reflect operational realities rather than arbitrary benchmarks. Third, implementing performance pressure “circuit breakers”—mechanisms that trigger resource replenishment interventions when pressure indicators exceed critical thresholds—can help prevent ethical lapses. Fourth, organizations should develop ethical decision frameworks specifically addressing common dilemmas arising from performance pressures in their particular context.

Additionally, our findings highlight the protective value of coworker support, suggesting organizations should cultivate supportive peer relationships through structured interventions. These include formal mentoring circles and peer coaching programs designed to facilitate both instrumental assistance and emotional reinforcement; collaborative team structures incentivizing resource sharing and joint problem-solving; comprehensive support skill training developing capabilities in providing effective feedback and emotional support; and formal recognition systems rewarding supportive behaviors. Implementation should follow a strategic sequence: first creating psychological safety through leadership modeling, then providing skills training and structural support, and finally reinforcing through recognition systems that institutionalize supportive behaviors as organizational norms.

While our findings emphasize the benefits of reducing performance pressure and enhancing coworker support, organizations must carefully consider potential unintended consequences of these interventions. Reducing performance pressure, if not strategically implemented, might inadvertently lower performance standards or lead to employee disengagement. Calibrating pressure reduction requires maintaining sufficient motivation while eliminating excessive strain. Organizations should establish optimal pressure thresholds that balance well-being with performance through regular monitoring and adjustment. Similarly, coworker support initiatives might create groupthink dynamics or diffuse accountability if not properly structured. To mitigate this risk, organizations should design peer support systems that simultaneously encourage constructive dissent and maintain individual accountability. Leaders should also remain vigilant for signs of collective moral disengagement within highly cohesive teams. The optimal implementation approach involves gradual adjustment of performance expectations coupled with clear communication about ethical standards, regular assessment of both performance metrics and ethical indicators, and structured accountability mechanisms that complement rather than undermine supportive peer relationships.

### 5.3. Limitations and Future Research Directions

Several limitations warrant consideration when interpreting our findings. First, our cross-sectional, single-source design raises methodological concerns. Although we conducted multiple statistical tests to address common method bias—including Harman’s single-factor test ([24]), confirmatory factor analysis with common latent factor, and sensitivity analyses with different control variable combinations—and found evidence mitigating these concerns, future research should employ multi-source data collection, time-lagged designs, or experimental methods to establish stronger causal relationships. Self-reported measures of sensitive constructs like UPB remain vulnerable to social desirability effects despite confidentiality assurances. Future research would benefit from methodological triangulation, including supervisor assessments, objective behavioral indicators, or experimental observations of unethical behavior.

Second, our sample characteristics—South Korean police officers in a hierarchical, predominantly male (94.29%), collectivist, public sector organization—limit generalizability. While this context provided an ideal setting for examining performance pressure, future research should test our model across diverse organizational settings, including private sector companies, less hierarchical structures, and more individualistic cultures, to determine whether our findings generalize beyond the specific cultural and organizational context studied.

Our theoretical framework suggests several promising directions for future research. Researchers might explore the comparative effectiveness of different boundary conditions in mitigating the performance pressure–UPB relationship, examining whether horizontal resources (peer support) and vertical resources (leader support) operate through distinct mechanisms or complement each other. Future studies should also investigate potential “dark side” effects of coworker support, exploring conditions under which strong peer cohesion might facilitate collective rationalization of unethical behavior through shared moral disengagement or create conformity pressures normalizing questionable practices. Additionally, researchers might examine threshold effects in the job strain–UPB relationship to determine whether critical levels of strain exist at which ethical decision-making fundamentally alters rather than deteriorates linearly.

Finally, incorporating physiological measures alongside self-reported measures would enable more comprehensive testing of our biopsychosocial conceptualization of job strain, potentially identifying biomarkers signaling elevated risk for ethical compromises. Future studies employing experience sampling methodologies could capture the dynamic unfolding of performance pressure, strain responses, and ethical decision-making over time, revealing how daily fluctuations affect ethical choices and distinguishing between chronic and acute effects of performance pressure.

## Figures and Tables

**Figure 1 behavsci-15-00573-f001:**
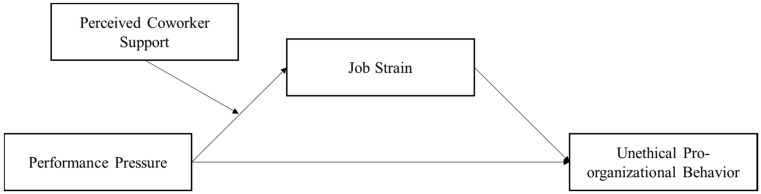
Hypothesized model.

**Figure 2 behavsci-15-00573-f002:**
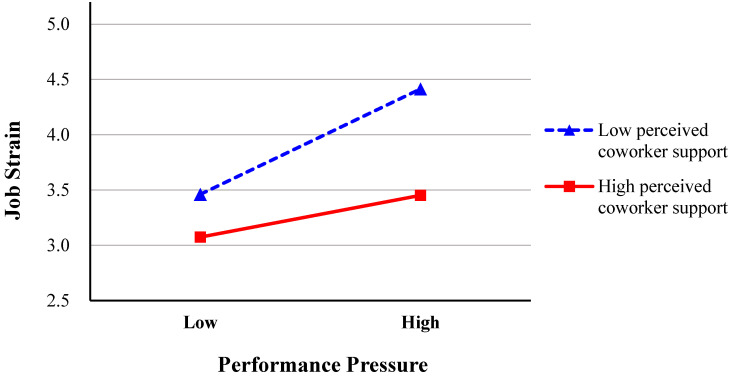
Moderating effect of perceived coworker support on the relationship between performance pressure and job strain.

**Figure 3 behavsci-15-00573-f003:**
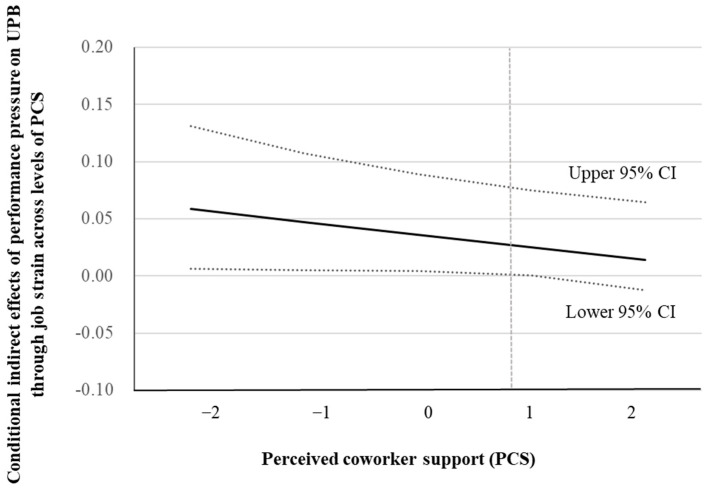
Conditional indirect effects of performance pressure on UPB through job strain across levels of perceived coworker support.

**Table 1 behavsci-15-00573-t001:** Descriptive statistics and correlations.

Variable		M	SD	1	2	3	4	5	6	7
1.	Age	35.55	8.46							
2.	Gender	1.06	0.23	−0.06						
3.	Education	1.66	0.62	0.16 **	0.11 *					
4.	Performance pressure	4.27	1.09	0.03	0.14 **	0.10 ^†^	(0.87)			
5.	Perceived coworker support	5.51	1.03	−0.11 *	−0.05	−0.02	−0.13 *	(0.94)		
6.	Job strain	3.62	1.34	0.02	0.04	0.08	0.27 ***	−0.28 ***	(0.89)	
7.	Unethical pro-organizational behaviors	3.37	1.17	−0.02	0.12 *	−0.05	0.26 ***	−0.08 ^†^	0.20 ***	(0.82)

Note: N = 280. Reliability is placed along the diagonal in parentheses. ^†^ *p* < 0.10; * *p* < 0.05; ** *p* < 0.01; *** *p* < 0.001 (two-tailed). 1. Age: year; 2. Gender: 0 = male, 1 = female; 3. Education: 1 = high school graduate, 2 = bachelor’s degree, 3 = graduate student, 4 = master’s degree and above, 5 = other.

**Table 2 behavsci-15-00573-t002:** Regression results for hypotheses: mediation.

	Job Strain		Unethical Pro-Organizational Behavior			
	M1	M2	M3	M4	M5	M6
1. Control variables						
Age	0.01	0.00	0.00	0.00	−0.01	−0.01
Gender	0.04	0.00	0.13 *	0.12 *	0.09	0.09
Education	0.07	0.05	−0.07	−0.08	−0.09	−0.09
2. Main effect						
Performance pressure		0.26 ***			0.26 ***	0.22 ***
3. Mediator						
Job strain				0.20 ***		0.14 *
Overall F	0.65	5.53 ***	1.79	4.19 **	6.25 ***	6.18 ***
R^2^	0.01	0.07	0.02	0.06	0.08	0.10
Change in F		20.03 ***		11.17 ***	19.26 ***	5.50 *
Change in R		0.07		0.04	0.02	0.02
Normal theory tests for indirect effect						
Sobel		Effect	SE	Z	*p*	
		0.040	0.020	2.038	0.042	
Bootstrap results for indirect effect						
Bootstrap		Effect	Boot SE	LL 95% CI	UL 95% CI	
		0.040	0.022	0.003	0.092	

Note: N = 280. * *p* < 0.05; ** *p* < 0.01; *** *p* < 0.001 (two-tailed). Bootstrap sample size = 10,000. SE = standard error; LL = lower limit; UL = upper limit; CI = confidence interval.

**Table 3 behavsci-15-00573-t003:** Regression results for hypotheses: main and moderating effects.

	Job Strain			
	M1	M2	M3	M4
1. Control variables				
Age	0.01	0.00	−0.02	−0.01
Gender	0.04	0.00	−0.01	−0.01
Education	0.07	0.05	0.05	0.05
2. Main effect				
Performance pressure		0.26 ***	0.23 ***	0.25 ***
Perceived coworker support (PCS)			−0.26 ***	−0.25 ***
3. Moderating effect				
Performance pressure × PCS				−0.13 *
Overall F	0.65	5.53 ***	8.83 ***	8.45 ***
R^2^	0.01	0.07	0.14	0.16
Change in F		20.03 ***	20.44 ***	5.77 *
Change in R		0.07	0.06	0.02

Note: N = 280. * *p* < 0.05; *** *p* < 0.001 (two-tailed).

**Table 4 behavsci-15-00573-t004:** Conditional indirect effects of performance pressure on UPB through job strain across levels of perceived coworker support.

		Unethical Pro-Organizational Behaviors			
Moderator	Level	Conditional Indirect Effect	SE	LL 95% CI	UL 95% CI
Perceived coworker support	Low	0.053	0.028	0.006	0.120
	Mean	0.037	0.021	0.005	0.091
	High	0.021	0.018	−0.002	0.072

Note: N = 280. Bootstrap sample size = 10,000. SE = standard error; LL = lower limit; UL = upper limit; CI = confidence interval.

## Data Availability

The raw data related to the variables included in the manuscript are available from the corresponding author upon reasonable request.

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
