# Peer review of "When Pressure Breeds Misconduct: Job Strain, Coworker Support, and Unethical Pro-Organizational Behavior"

_behavsci, 2025, doi:10.3390/bs15050573_

Round 1
Reviewer 1 Report
Comments and Suggestions for Authors
Thank you for your work on this manuscript, which addresses an important topic in the study of unethical pro-organizational behavior. I recommend a major revision, as several key aspects require further refinement to enhance the theoretical contribution, methodological rigor, and practical implications. My comments are somewhat detailed, as I have aimed to be as precise as possible to support you in strengthening the manuscript. I appreciate your efforts in considering these revisions.

Reviewer 2 Report
Comments and Suggestions for Authors
Thank you for the opportunity to review this manuscript.
This article is a valuable contribution to the fields of organizational behavior and business ethics. It addresses a timely and important issue on how performance pressure can lead to unethical UPB, and how coworker support may buffer this effect. The authors bridge theoretical gaps by integrating Conservation of Resources (COR) theory with a moderated mediation model in both conceptual and practical relevance.
Strengths include 1. Theoretical Grounding. The application of COR theory provides a solid and well-established psychological framework for understanding the mechanisms linking performance pressure and unethical behavior. The incorporation of job strain as a mediator and perceived coworker support as a moderator demonstrates thoughtful model development and theoretical integration. 2. The study employs validated multi-item scales with strong reliability (Cronbach’s α > .80) and uses appropriate statistical methods, including confirmatory factor analysis, bootstrapping, and moderation/mediation modeling. The sample size (N = 280) is adequate for the complexity of the analysis, and the use of parceling techniques is appropriate for managing model complexity. 3. Practical Implications.
The findings offer actionable insights for organizations seeking to mitigate UPB by cultivating supportive work climates. The study reinforces the dual nature of performance pressure—both as a motivator and as a risk factor for ethical compromise—and highlights coworker support as a practical, internal buffer.
While the manuscript is well written, several areas could benefit from further clarification: 1. The exclusive use of self-reported, cross-sectional data introduces the possibility of common method variance (CMV) and social desirability bias, particularly given the sensitive nature of UPB. While Harman’s single-factor test is reported, this method alone is insufficient. Future studies could strengthen causal inferences by incorporating multi-source data or time-lagged designs. 2. The sample is relatively homogeneous—primarily male (94%) and drawn from a single occupational group (South Korean police officers). This limits the external validity of the findings, especially in different cultural contexts or private-sector environments where UPB may manifest differently. 3. While the study acknowledges the ethically questionable nature of UPB, it does not fully explore the moral ambiguity or potentially prosocial intentions underlying such behavior. A deeper engagement with ethical theory (e.g., moral disengagement, ethical climates, or values-based leadership) would enrich the conceptual framework.
Comments on the Quality of English Language
The English is sufficient for publication, especially given the technical nature of the work. However, a light professional copyediting would improve the writing from “clear but clunky” to smooth and polished, increasing the paper’s impact. For example, in Introduction (lines 30–33): “UPB… actions intended to benefit the organization or its members, but which violate societal values, laws, or standards... they threaten long-term organizational sustainability…” Since UPB is defined early in the manuscript, later references should build upon rather than restate the definition. The focus could shift to elaborating consequences or ethical tensions instead of reiterating its basic nature.
Round 2
Reviewer 1 Report
Comments and Suggestions for Authors
Dear Authors,
Thank you for the significant efforts you have made to revise your manuscript. The revised version demonstrates a clear improvement in theoretical precision, methodological rigor, and practical relevance.
The repositioning of job strain as a biopsychosocial mediator represents a meaningful contribution to the literature on unethical pro-organizational behavior. Your discussion now clearly distinguishes this mechanism from previously studied cognitive explanations such as moral disengagement, self-regulation failure, and cognitive depletion. This clarification helps to position your work as a novel and integrated contribution to the ethical decision-making literature.
Your justification for the moderating role of coworker support is also more persuasive. By contrasting this horizontal interpersonal mechanism with more traditionally studied vertical factors such as ethical leadership or organizational justice, you strengthen the originality and theoretical relevance of your model.
On the methodological side, your expanded efforts to address common method bias are welcome. The inclusion of a CFA with a common latent factor, the use of ULMC techniques, and your discussion of the attenuation of interaction effects all enhance the credibility of your results. Your explanation of the selection of control variables and your sensitivity analyses to address the risk of misinterpreting their effects are also thoughtful and well executed.
Finally, the practical recommendations proposed in your discussion section are now more concrete and actionable. The description of implementation strategies, particularly regarding how to foster coworker support, adds value to the paper for practitioners.
That said, I would recommend a minor revision on one specific point. Your paper emphasizes the benefits of reducing performance pressure and enhancing coworker support, but it does not yet consider the potential unintended consequences of such interventions. For example, while reducing pressure can prevent ethical lapses, it may also risk lowering performance standards or lead to disengagement if not well calibrated. A brief paragraph addressing this trade-off would enrich the discussion and reflect the complexity of translating these findings into practice.
Again, I appreciate the clarity and depth of your revisions. I believe this paper makes a strong contribution and will be ready for publication following this minor adjustment.
Sincerely,
Reviewer 1
